# A unified analysis of evolutionary and population constraint in protein domains highlights structural features and pathogenic sites
Stuart A. MacGowan [1], Fábio Madeira [1,2], Thiago Britto-Borges[1,3] & Geoffrey J. Barton [1] ✉

Protein evolution is constrained by structure and function, creating patterns in residue conservation that are routinely exploited to predict structure and other features. Similar constraints should affect variation across individuals, but it is only with the growth of human population sequencing that this has been tested at scale. Now, human population constraint has established applications in pathogenicity prediction, but it has not yet been explored for structural inference. Here, we map 2.4 million population variants to 5885 protein families and quantify residue-level constraint with a new Missense Enrichment Score (MES). Analysis of 61,214 structures from the PDB spanning 3661 families shows that missense depleted sites are enriched in buried residues or those involved in small-molecule or protein binding. MES is complementary to evolutionary conservation and a combined analysis allows a new classification of residues according to a conservation plane. This approach finds functional residues that are evolutionarily diverse, which can be related to specificity, as well as family-wide conserved sites that are critical for folding or function. We also find a possible contrast between lethal and non-lethal pathogenic sites, and a surprising clinical variant hot spot at a subset of missense enriched positions.

Comparative analysis of homologous proteins and their three-dimensional structures has systematically established that sequence is constrained by structure and function[1,2]. This classic result is fundamental to our understanding of molecular evolution and underlies the development of methods to predict protein structure[3–10] and to interpret human genomes[11–15]. Key elements of the sequence-structure-function relationship are that buried residues evolve more slowly than exposed residues[16,17], residues at interfaces tend to conserve physicochemical properties[18–21] and co-varying residues are often close in space[9]. These trends, together with secondary structure[3] and other positional physicochemical preferences[22], are among the patterns that AI algorithms like AlphaFold[23] can exploit and are therefore crucial to their success.

The same structural constraints that mark evolutionary variation were also expected to influence population polymorphisms[24]. However, systematic confirmation of the effect had to wait until resequencing projects[25,26] sampled enough variation to enable the analysis. Since then the distribution of missense variants in humans was shown to be strongly influenced by protein structure, with features like core residues, catalytic sites, and interfaces showing evidence of constraint in aggregate[27–29]. In parallel, several methods emerged to calculate population constraint at different scales and were consistently found to be useful for variant pathogenicity prediction[25,30–34]. However, despite this progress, the application of population constraint for structural inference is less well-developed.

One challenge is that structural constraints in proteins are inferred from residue-level conservation patterns and so most existing constraint scores, which apply to whole genes or domains, are too coarse for this task. However, residue-level resolution is possible with the aid of protein family

[1]Division of Computational Biology School of Life Sciences University of Dundee, Dow Street Dundee, DD1 5EH Scotland, UK. [2]Present address: European Bioinformatics Institute (EMBL-EBI), Wellcome Trust Genome Campus, Hinxton, Cambridge CB10 1SD, UK. [3]Present address: Section of Bioinformatics and Systems Cardiology, Department of Internal Medicine III and Klaus Tschira Institute for Integrative Computational Cardiology, Heidelberg University Hospital, Heidelberg, Germany. ✉e-mail: gjbarton@dundee.ac.uk

sequence alignments where the signal can be enhanced by combining variation across homologous sites. This approach previously enabled the detection of low-frequency cancer driver mutations[35–37] and clusters of pathogenic mutations in domain families[38], while population variant distributions in pseudo-paralogous alignments were shown to highlight a range of genomic features, including start codons, 5'-UTRs, CDS regions, and wobble nucleotides[26]. Since then, population constraint computed in this way has also proven useful for pathogenicity prediction[39–43], and its association with structural features has been observed in specific proteins[39,44].

The capacity to calculate population constraint at the residue-level, presents a unique opportunity to conduct a comparative analysis of population and evolutionary constraint in proteins. At the outset, we can consider the essential differences between the underlying datasets. The diversity catalogued in databases like Pfam[45] reflects the cumulative effects of hundreds of millions of years of evolution, shaped by spatial and temporal scales that give rise to an enormous potential for variation and selection. In contrast, population datasets like gnomAD[46] capture genetic diversity influenced by more recent events, migrations and drift, all within the genomic context of a single species. Given these profound differences, the question arises whether the signatures of population constraint within humans offer a distinct perspective from the deep evolutionary conservation observed across species. Theoretical frameworks, such as the McDonald-Kreitman test[47], have exploited such contrasts to discern selection pressures, underscoring the potential of an integrated approach, but to our knowledge, it has never been applied at as fine a resolution as single amino acids.

In this work, we develop a new residue-level population constraint metric computed over the columns of protein family alignments with variants from gnomAD[48]. We apply this method to classify residues in thousands of protein families from Pfam[45], and assess their properties with respect to features from experimentally determined protein three-dimensional structures[48,49], including solvent accessibility, protein-protein and protein-ligand interfaces and pathogenic variants from ClinVar[50]. We then combine our population constraint score with a conventional measure of evolutionary conservation, based on residue diversity in protein family alignments, to reveal the structural and functional properties of a new classification of residues.

## Results

### A residue level map of population constraint in the human proteome

**The distribution of population missense variants amongst humans is correlated with evolutionary conservation in protein domain families.** We mapped 2.4 million population missense variants from gnomAD[46] to 1.2 million positions within 5,885 protein domain families from Pfam[45]. This covers 5.2 million residues of the human proteome. Figure 1 shows a schematic of a variant annotated alignment (Fig. 1a) alongside illustrations of the distribution of variants within domains in different contexts (Fig. 1b–e; Supplementary Fig. 1). In any given protein, most residues are not variable at all in gnomAD and the observed missense variants are sparsely distributed across the sequence (Fig. 1b). As a result, the distribution of missense variants along a sequence provides limited information about constraint at individual residues, which is why existing methods average over larger windows, such as continuous ranges of linear sequence[31,33] or volumes in 3D space[34,51].

An alternative is to collate variation that occurs at homologous positions in protein domains as embodied by the columns of multiple sequence alignments. Sites within Pfam[45] domains containing many human paralogs have more missense variants (Fig. 1c), and so we hypothesised that a comparative analysis of variants at homologous positions could yield a residue-level resolution of population constraint. To test this hypothesis, we compared the domain distribution of missense variants to Shenkin's measure of evolutionary conservation[52]. Figure 1d shows this relationship in the SH2 domain family and highlights the strong positive correlation between population missense variants and evolutionary conservation such that more population variation is observed at positions with greater evolutionary

diversity. This behaviour is also observed in many other domains where we detected a significant positive association in 900 Pfam domains, suggesting that this is a general phenomenon (Fig. 1e). In contrast, synonymous variants are not associated with evolutionary diversity, which serves as a convenient negative control. This result suggests that population variants are broadly constrained by the same features that constrain evolutionary substitutions. For example, in the SH2 domain, evolutionary conservation and population constraint are both indicative of structural constraints that can be observed in protein structures, including inter-domain interaction sites on the SH2 surface, as shown in Fig. 1f. We find a significant relationship in 85% of the 140 families with more than 100 human paralogs, compared to 12% among those with fewer human sequences, emphasising the dependency on the number of human paralogs to observe this signal.

**A Missense Enrichment Score (MES) to represent population constraint in protein domains.** To compare population constraint between different domains that vary in depth, we developed the Missense Enrichment Score (MES), which quantifies relative population constraint at each site in a domain. MES has two components: the odds ratio of the position's rate of missense variation compared to the rest of the domain (MES), and a $p$ value indicating the significance of the deviation of MES from 1 (see Methods). This is a good formulation for structural analyses across multiple families because it is intrinsically normalised for varying numbers of human paralogs, as well as any heterogeneity in overall constraint due to factors like gene essentiality. The $p$ value is calculated with Fisher's exact test and so holds no distributional assumptions and is appropriate even for small families where variant counts are low. The $p$ value is sensitive to the overall variant counts in the family, affected by the number of human paralogs and their coverage in gnomAD.

From the MES, we define missense-depleted positions as those with a lower rate of missense variation than other positions in the domain at a specified, two-tailed $p$-value threshold (i.e., MES < 1; $p < 0.1$), which are sites that are under relatively strong constraint. We similarly define missense enriched (MES > 1; $p < 0.1$) and missense neutral positions ($p \geq 0.1$), which represent different grades of polymorphic positions. We found 5086 positions in 766 families met this threshold of missense depletion (covering 365,300 residues in the human proteome), while 13,490 positions in 3591 families were enriched in missense variants (340,829 residues in human). In the following, we describe the structural and functional features of these positions with reference to 61,214 three-dimensional structures from the Protein Data Bank[49], covering 40,394 sequences from 3661 Pfam domains, and over 10,000 variants labelled pathogenic in ClinVar[50]. We also compare their properties to comparable sets of evolutionarily conserved and unconserved residues, defined with their Shenkin diversity (see Methods).

**Population-constrained sites are enriched in buried residues and binding-sites.** Population constraint is strongly associated with solvent exposure ($\chi^2 = 1285$, df = 4, $p \approx 0$, $n = 105,385$; Supplementary Table 1; Supplementary Fig. 2), such that missense-depleted sites are predominantly buried in the protein core, while missense-enriched sites tend to be found at surface-exposed positions. We also observe that missense-depleted positions are enriched for residues interacting with proteins or ligands (Fig. 2). The effect is particularly pronounced for ligand binding sites, whereas for protein-protein interfaces, it is detected only when considering solvent accessibility due to the infrequent interactions of buried residues with other proteins, emphasising the importance of this control.

For both ligand binding-sites and protein-protein interfaces (Fig. 2a, b), the effects are greatest at surface sites, where residues encounter fewer constraints from packing and folding, making external interactions more relevant. These patterns align closely with observations at evolutionarily conserved sites, implying that the constraint captured by MES could be useful to predict these features. In many cases, the feature enrichments are similar for population-constrained and evolutionary-conserved sites, even though only human paralogs contribute to the MES calculation. This suggests that MES may be sensitive to a wider range of binding-sites,

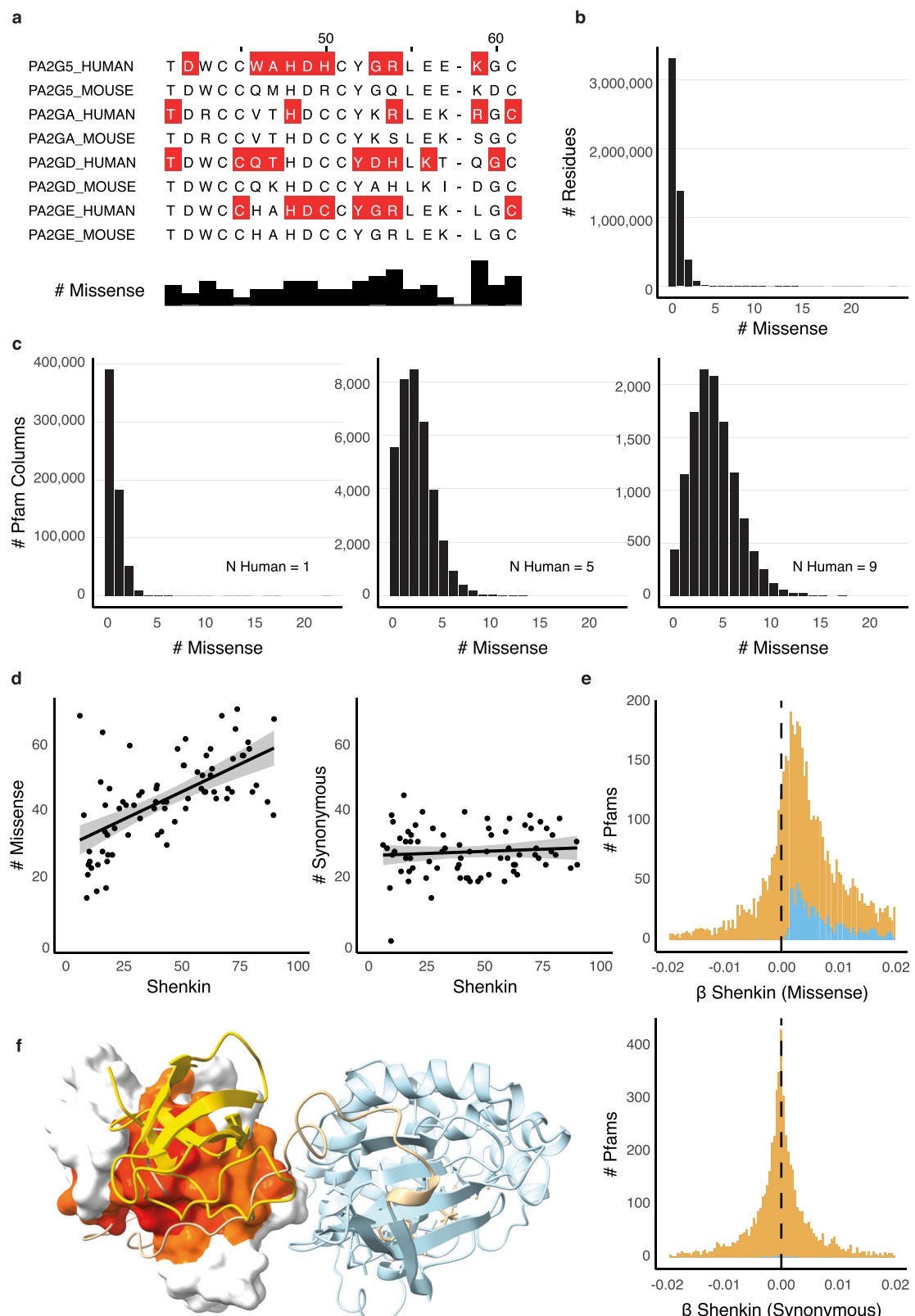

including those at evolutionarily diverse positions, a hypothesis that we revisit later.

**A complex distribution of pathogenic variants with respect to constraint and solvent accessibility.** We also tested whether pathogenic

variants were enriched within population-constrained positions (Fig. 2c). Overall, ClinVar[50] pathogenic missense variants are enriched within missense-depleted sites. However, there is a complex pattern to the distribution of pathogenic variants in different solvent accessible regions. While variants at missense-depleted sites among surface positions are

**Fig. 1 | Population diversity in Pfam domains is often positively correlated with evolutionary diversity. a** An excerpt of the Phospholipase A2 1 domain (PF00068) highlighting sites with missense variants from gnomAD (red). The histogram shows the number of missense variants at each position across all 10 human sequences in the Pfam domain. **b** Frequency distribution of gnomAD missense variants across all amino acid residues in Pfam domains ($n = 5,162,957$ residues). **c** Frequency distributions of gnomAD missense variants over alignment columns of Pfam domains containing 1, 5 or 9 human sequences ($n = 635,974$, 36,317, and 12,020 sites). **d** The total number of gnomAD missense or synonymous variants vs. the Shenkin diversity at each position across SH2 domains (PF00017) and linear regressions (left panel:

$m = 0.33$, $p = 6.45 \times 10^{-9}$; $c = 28$ $p \approx 0$; right panel: $m = 0.00$, $p = 0.46$; $c = 20$, $p \approx 0$; both $n = 75$ sites). **e** The distribution of regression coefficients for gnomAD missense or synonymous variant totals against Shenkin divergence across Pfam ($n = 5975$ domains). Regression coefficients with $p < 0.05$ are coloured blue. **f** Inter-domain interactions of the SH2 domain in inactivated Proto-oncogene tyrosine-protein kinase Src (SRC; PDB ID: 2src[76]). The surface is coloured according to the Missense Enrichment Score (MES; red to yellow) calculated from the SH2 Pfam domain (PF00017). Figure created with Jalview[77] and UCSF Chimera[78]. Additional dataset statistics are shown in Supplementary Fig. 1.

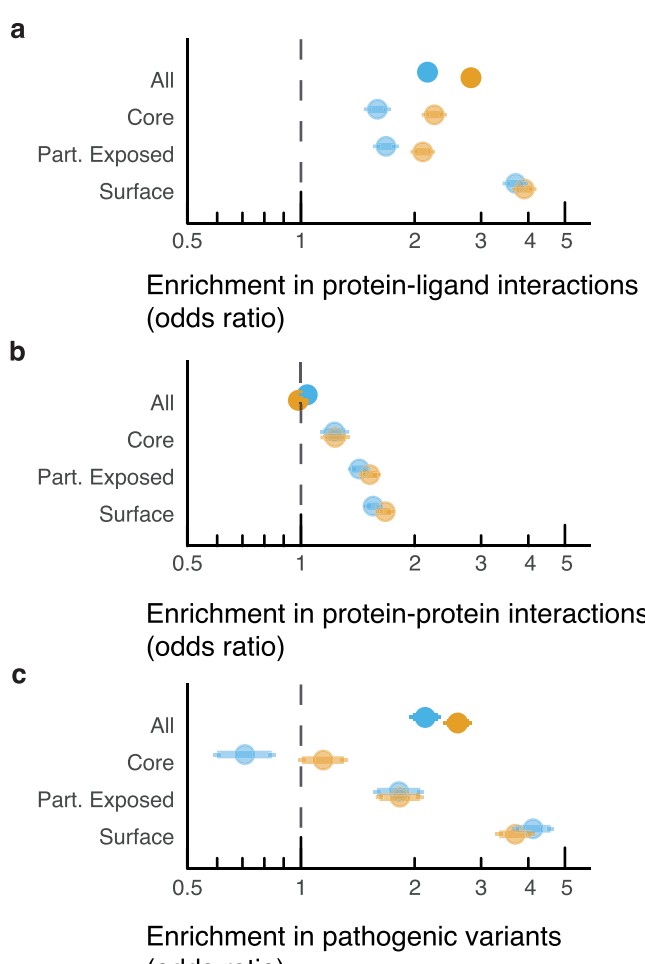

**a**

Enrichment in protein-ligand interactions (odds ratio)

**b**

Enrichment in protein-protein interactions (odds ratio)

**c**

Enrichment in pathogenic variants (odds ratio)

**Fig. 2 | The structural and functional properties of missense depleted positions compared to evolutionarily conserved positions.** Odds ratios (95% CI) measuring the enrichment of **a** protein-ligand interactions, **b** protein-protein interactions and, **c** pathogenic variants within conserved (orange) or missense depleted columns (blue) and stratified by solvent exposure class. See Supplementary Fig. 3 for comparison to missense enriched and evolutionarily divergent positions.

## A unified analysis of population constraint and evolutionary conservation

**Categorising residues by population and evolutionary variation.** We devised an approach to integrate signals from population and evolutionary variation by subclassifying conserved and unconserved positions by MES (Fig. 3). This scheme yields four primary categories corresponding to the quadrants of a conservation plane (Fig. 3a). These are conserved-missense depleted (CMD), conserved-missense enriched (CME), unconserved-missense depleted (UMD) and unconserved-missense enriched (UME). Among 109,790 sites from the 605 Pfam families with at least one missense depleted site, the most common categories were CMD (3% of sites; 10% of residues) and UME sites (2%; 10%), which show consistency between evolutionary conservation and population constraint, while the discordant CME (2%; 2%) and UMD categories (1%; 2%) were rarer (Fig. 3b). This higher prevalence of CMDs and UMEs over CMEs and UMDs reflects the overall correlation between evolutionary conservation and the missense distribution described earlier, but despite this trend there are still over 100,000 residues of the human proteome within the uncommon UMD and CME categories (Supplementary Table 2).

Comparing the functional properties of these classes reveals that population constraint stratifies conserved and unconserved sites, where missense-depleted positions are usually enriched in functional residues (Fig. 3c–f). For example, among surface sites, CMDs are exceptionally enriched in ligand interactions (OR = 5.0, $p \approx 0$), but CMEs are not (OR = 1.0, $p = 0.77$), and there is a similar trend comparing UMD and UME positions (Fig. 3d). The effect can be so great that population constraint supersedes conservation, such as how protein interface residues are depleted among CMEs (OR = 0.88, $p = 8.2 \times 10^{-11}$), but enriched at UMDs (OR = 1.1, $p = 2.7 \times 10^{-9}$; Fig. 3e). These patterns are broadly reinforced when considering pathogenic variants (Fig. 3f), and in partially exposed sites (Supplementary Fig. 4).

**Population constraint at unconserved sites (UMDs) indicates potential specificity determining positions.** Unconserved sites are often overlooked since it is hard to interpret their significance, but the diversity at these sites can be the key to protein specificity. This raises the prospect that population constraint at unconserved sites is a characteristic of specificity-determining positions (SDPs), which are sites that impart ligand or substrate specificity to the domain. Supporting this idea, UMDs are prominent in the GPCR (Fig. 4a–d) and nuclear receptor protein families (Fig. 4e–h), where several surround their highly variable receptor binding pockets, regions clearly critical for ligand specificity. In these families, UMD sites are the most enriched for ligand binding residues (Fig. 4d and h), which reflects the fact that cognate ligands vary across these proteins. We also found similar results in the Ankyrin repeats where UMD-like sites were involved in functional interactions[44]. While these examples illustrate the nature of UMDs, the consistent enrichment of ligand interactions, protein-protein interfaces, and pathogenic variants at UMD sites supports the conclusion that it is a general phenomenon (Fig. 3 and Supplementary Table 3).

**Population constraint at evolutionary conserved sites (CMDs) is a strong indicator of structural or functional importance.** Conserved sites across a family are typically recognised as important to folding and

over four times more likely to be pathogenic (OR = 4.1, $p \approx 0$), among core positions missense depleted sites are less likely to harbour reported pathogenic variants (OR = 0.7, $p = 0.00002$). The effect is so great that missense-depleted sites on protein surfaces have a higher proportion of pathogenic variants to population variants (388 vs. 27,420; 1.4%), than those in buried positions (155 vs. 27,504; 0.6%). A similar trend is observed with respect to evolutionary conservation. These surprising results could indicate that deleterious mutations at the most essential sites, such as critical core positions, are more likely to be genetically lethal and thereby depleted in the sort of clinical pathogenic variants reported in ClinVar.

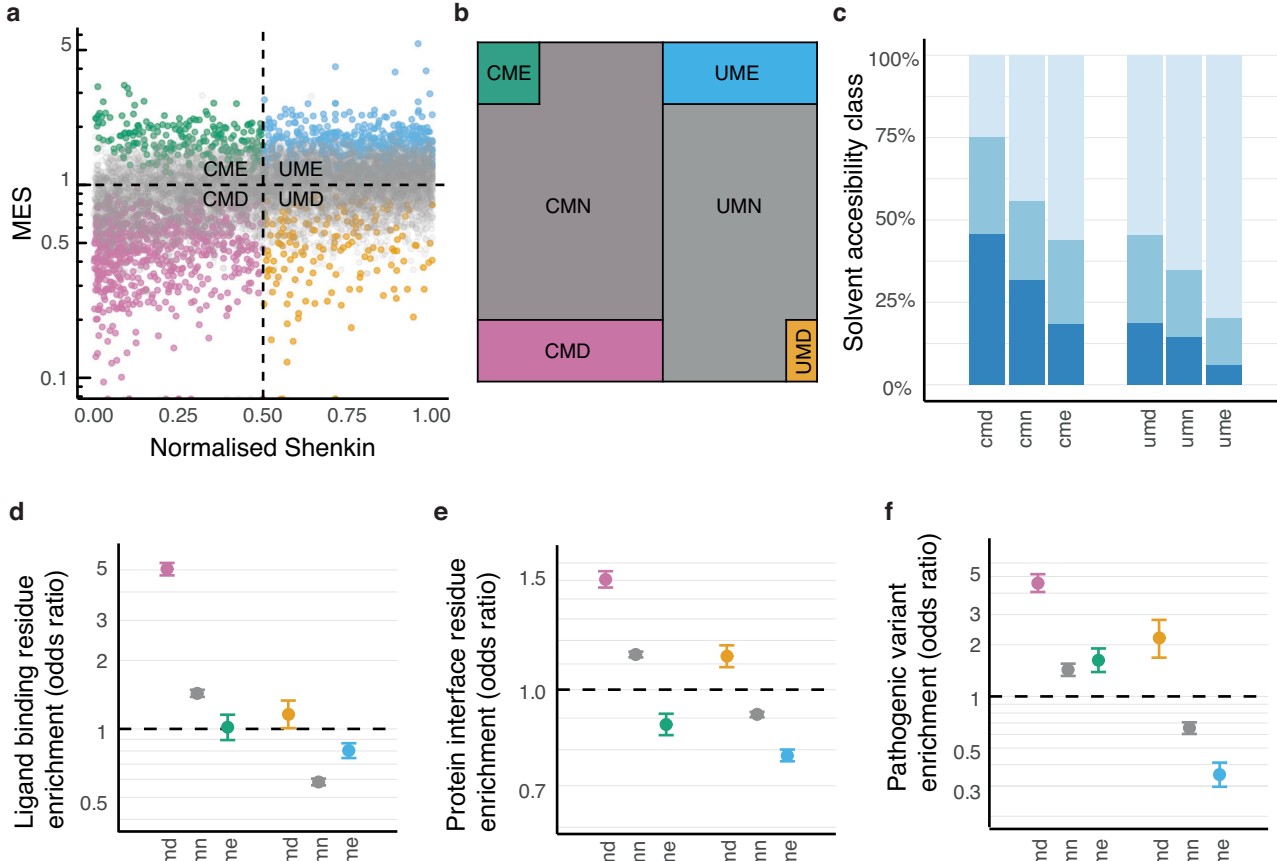

**Fig. 3 | Classifying sites in protein domains with evolutionary conservation and population constraint. a** The missense enrichment score (MES) vs. normalised Shenkin diversity for a random sample of 10,000 sites from 282 Pfam domains. The quadrants of this conservation plane correspond to sites that are conserved-missense depleted (CMD; pink), conserved-missense enriched (CME; green), unconserved-missense enriched (UME; blue) and unconserved-missense depleted (UMD; orange). Missense neutral sites where MES $p < 0.1$ are coloured grey. Points on the x-axis represent sites with no missense variants (i.e., MES = 0). **b** Area plot depicting the relative proportion of residues in each category. Note that the main categories have been positioned to the corners of the plot for clarity. **c** The proportions of site solvent exposure class across different constraint classes (dark to light blue corresponds to core, partially exposed and surface). **d–f** The enrichment of protein-ligand binding residues, protein interacting residues and ClinVar[50] pathogenic variants amongst surface sites in each category. Error bars indicate 95% confidence intervals. Supplementary Table 3 presents related data focussed on comparing UMD and UME positions.

function, but it appears that differences in population constraint among them provides further discrimination. CMDs are predominantly found at buried sites, with almost 75% corresponding to core or partially exposed positions (Fig. 3c), emphasising their importance to structural maintenance. Where CMDs are found outside the core, they are the most highly enriched in interactions (Fig. 3d, e), with ligand binding residues 5-fold enriched (OR = 5.0, $p \approx 0$) and protein binding residues 1.5-fold enriched (OR = 1.5, $p \approx 0$) among surface sites, and similar trends found at partially exposed residues (Supplementary Fig. 4), emphasizing the role of interactions where packing constraints are reduced. A substantial enrichment of pathogenic variants within CMDs at surface (OR = 4.6, $p \approx 0$) and partially exposed sites (OR = 2.2, $p \approx 0$) underscores their importance. The unexpected absence of an enrichment of pathogenic variants at core CMDs (OR = 0.85, $p = 0.06$) is discussed later.

In our case-studies, CMDs are clustered around features that are constant across all member domains. In GPCRs, this includes the allosteric sodium binding site and the tryptophan toggle switch, which play pivotal roles in the universal activation dynamics common across class A GPCRs (Fig. 4a). In nuclear receptors, CMDs are found in the coactivator binding cleft, including the important charge-clamp residue[53], while others are packed tightly at the interhelical interface bridging the coactivator and hormone pocket, a region likely important for folding as well as coactivator and hormone cooperativity[54] (Fig. 4e). Finally, outside this study we identified CMD sites as the critical folding residues in Ankyrin repeats[44].

**Pathogenic variant hotspots and putative genetically lethal sites**. CMDs at core sites were not enriched in pathogenic variants (OR = 0.85, $p = 0.06$), despite substantial effects at surface (OR = 4.6, $p \approx 0$) and partially exposed residues (OR = 2.2, $p \approx 0$). This pattern echoes the results for population and evolutionarily constrained sites when considered in isolation (Fig. 2c). However, the combined analysis isolates this effect specifically to CMDs, the most highly constrained sites, distinguishing them from CMEs and UMDs, which do not exhibit this unique behaviour (Supplementary Fig. 4). This distinction refines our earlier hypothesis: the lower prevalence of pathogenic variants in buried CMDs is possibly related to the risk of genetic lethality, coupled with a sampling bias in ClinVar. Buried CMDs are likely to be critical for structural integrity or functional activity, with tight restrictions on permissible amino acid substitutions. In contrast, although UMDs are implicated in protein specificity, numerous substitutions between homologs at these sites suggests an element of structural plasticity. Similarly, the accommodation of a variety of missense alleles in CMEs means that genetically lethal variants must be rare at these sites.

Focusing on CMEs, we observe a consistent enrichment of pathogenic variants across all exposure levels, including the protein core (OR = 2.4, $p \approx 0$; Supplementary Fig. 4). This result significantly diverges from previously reported trends, which show an anti-correlation between the locations of population and clinical variants[28,29,55]. Moreover, this observation suggests that pathogenic variants at CME sites are a significant contributor

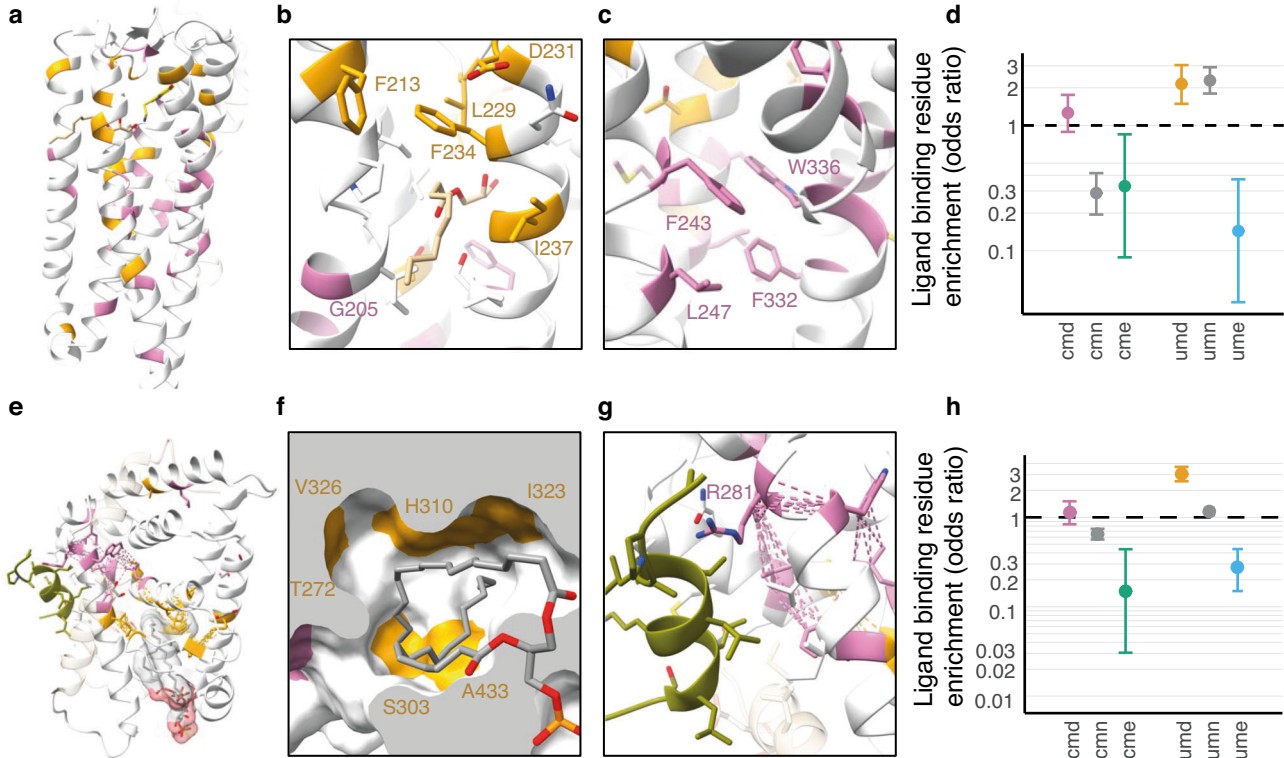

**Fig. 4 | The contrasting roles of unconserved-missense depleted (UMD) vs. conserved-missense depleted (CMD) sites in the Class A Rhodopsin GPCRs and nuclear receptors.** UMD (gold) and CMD (pink) sites in **a** 5-hydroxytryptamine receptor 2 A (5-HT$_{2A}$) bound to serotonin (yellow) and monoolein (tan) in the GPCRdb[79] refined model derived from PDB ID: 7wc4[80]. **b** The side-extended pocket of 5-HT$_{2A}$ with bound monoolein. Residues within 5 Å of monoolein are shown as sticks. **c** Conformational changes in 5-HT$_{2A}$ upon receptor activation in the vicinity of the W336 toggle position. The activated state in the GPCRdb refined PDB ID: 6wha[81]. **d** Enrichment of ligand binding residues amongst sites in Class A Rhodopsin

GPCRs. **e** Steroidogenic factor 1 (STF1) bound to phosphoinositide PIP$_2$ and the coactivator peptide PGC1-α (olive) in PDB ID: 4qk4[54]. **f** the lipid hydrophobic tails of PIP$_2$ bound within the hormone pocket showing close interactions with UMD positions that contribute to the interior surface of the pocket in this receptor. **g** The coactivator binding groove includes the CMD residue R281 located toward the C-terminal end of H3, which is part of the CMD cluster. **h** Enrichment of ligand binding residues amongst sites in the nuclear receptor ligand binding domain Pfam (PF00104). Structural illustrations created with Jalview[77] and UCSF Chimera[78].

to the documented enrichment of pathogenic variants within core residues[28,55]. The identification of CMEs as a pathogenic variant hotspot therefore marks a significant departure from established patterns and invites further investigations of the underlying mechanisms driving pathogenicity at these sites.

Finally, the UMDs we suggest are implicated in specificity, are another pathogenic variant hotspot. They are significantly enriched in pathogenic variants among surface and partially exposed sites (OR = 2.2, $p = 2.7 \times 10^{-8}$ and OR = 2.2, $p \approx 0$), and despite high uncertainty, are significantly more likely to harbour pathogenic variants than UME sites in the core (OR = 0.90–2.5 vs. 0.14 − 0.65 [95% confidence intervals]; Supplementary Fig. 4). Some of the relative differences in pathogenic variant enrichment between UMD and UME sites are exceptional (Supplementary Table 3). For example, surface UMDs are 6.5-fold enriched in pathogenic variants compared to UMEs (OR = 6.5, $p \approx 0$). These data are consistent with our characterisation of UMD sites as specificity determinants, but they also highlight the increased relevance of population constraint to clinical variation.

## Discussion
In this study, we have explored the interplay between population and evolutionary variation within protein families, establishing a nuanced understanding of how these forces reflect protein structure and function. Our results that missense depleted residues are enriched in structural and functional features demonstrate that population constraint can be exploited for structural inference. We have shown that an integrated analysis of evolutionary and population constraint discriminates sites

with distinct functional properties. We have defined new constraint categories that not only highlight functional and pathogenic sites, but also distinguished residues conferring protein specificity (UMDs) from those maintaining family-wide attributes (CMDs). When further integrated with residue solvent exposure, we also found sites where deleterious variants may pose a greater risk of genetic lethality (buried CMDs) and uncovered several pathogenic variant hotspots (CMEs, UMDs and exposed CMDs).

To our knowledge, the only existing framework that integrates population and evolutionary variation is the McDonald-Kreitman test[47], which is widely used to detect selection effects at gene-level. This test compares the ratios of non-synonymous to synonymous substitutions between species ($D = Dn/Ds$), to the within-species ratio ($P = Pn/Ps$), where $D > P$ indicates positive selection, $D < P$ implies negative selection and $D = P$ suggests neutrality. Although the test is defined with respect to substitutions and polymorphisms in two species, we can compare these states to our residue categorisation. Accordingly, unconserved-missense depleted sites (UMDs), which diverge across homologs but are constrained in the population, are aligned with positive selection, which has interesting implications for the evolution of protein specificity. Meanwhile, conserved-missense enriched sites (CMEs) are aligned with negative selection, in keeping with the high prevalence of pathogenic variants at these sites. Taking the diagonal of the conservation plane (Fig. 3a) to approximate $D = P$, conserved-missense depleted (CMDs) and unconserved-missense enriched sites (UMEs) would be interpreted as neutral. However, although this appears a reasonable label for UMEs, which are consistently devoid of functional features, it is poor description of CMDs, which are clearly under the strongest constraint. This

breakdown of an otherwise logical correspondence suggests a possible advantage of our approach.

The strength of our method to highlight important sites among unconserved positions could overlap with the signal from evolutionary covariation or other alignment-based methods, such as phylogeny-aware approaches that exploit conservation in orthologous and paralogous groups. In preliminary work we found no systematic relationship between our constraint categories and EVcouplings scores[56] in the nuclear receptor ligand binding domains (Supplementary Data 1). We also found little overlap between UMD sites and specificity-determining positions (SDPs) predicted by SDPfox[57] (Jaccard similarity = 0.02, between 1902 SDPfox predicted SDPs and 604 UMDs, across 130 Pfam domains). These initial comparisons suggest that our approach is complementary to existing methods.

Given the pervasive application of evolutionary conservation in biology, we expect our missense enrichment score will have broad relevance to a variety of research communities; we have ourselves applied MES to help characterise ligand binding sites[58] and for a detailed analysis of Ankyrin repeats[44]. Residue-level population constraint presents an opportunity to augment the evolutionary data now routinely fed to models developed for variant pathogenicity prediction, functional site identification and structure prediction. Modern machine learning methods can exploit the complex relationships we have revealed between population constraint, evolutionary conservation, residue burial, interactions and pathogenicity, and our results could help to guide appropriate implementations to this end. Models designed to exploit these patterns could be superior at recognising functional or pathogenic unconserved residues, successful at distinguishing specificity-determining sites or flagging potentially lethal variants. As population datasets continue to grow, this could be particularly helpful is for proteins with limited homology that generate shallow alignments, especially when data for more species become available.

## Methods
### Mapping population variants to Pfam alignments (Pfam-gnomAD)
Protein family alignments were downloaded from Pfam (v31)[59,60] and parsed using Biopython (v1.66, with patches #768 #769)[61]. Alignments with at least one human sequence were filtered to exclude TrEMBL sequences and leave the higher confidence SwissProt sequences. This yielded a set of 6042 alignments containing 392,703 sequences—including 44,530 human sequences—that we will refer to as PfamSP.

Population variants from the gnomAD Version 2 dataset were mapped to the human sequences in PfamSP. For each sequence in an alignment, genomic mappings were retrieved from Ensembl[62,63] via the Ensembl API and variants in these regions were loaded from a local copy of the gnomAD VCF with PyVCF. The PyVCF variant records were parsed into a Pandas DataFrame and UniProt residue mappings were derived from the Ensembl VEP SwissProt and Protein_position annotations. We provide a schematic of the procedure in Supplementary Fig. 5. In this way, we could link each variant in gnomAD to a column in PfamSP. This gave 2,405,900 missense variants from gnomAD mapped to 5,194,362 residues within 1,188,133 columns in 5985 PfamSP domains.

### Mapping clinical variants to Pfam alignments
The same procedure outlined above was used to map ClinVar[50] variants to PfamSP from a VEP-annotated ClinVar VCF.

### Pfam alignment conservation
Conservation scores for the alignments in PfamSP were calculated with AACons[64]. AACons provides 18 different conservation scores spanning identity-based (e.g., Shenkin[52]), physicochemical-based (e.g., Taylor[65], Zvelebil[3], redundancy weighted (e.g., Valdar[66]) and others and all scores are present in our provided datasets. In this analysis, we opted to use the Shenkin score to represent evolutionary conservation due to its simple

interpretation as reflecting the amino acid diversity at each site. This score has also proven effective in our previous work on Ankyrins[44].

Throughout the text we refer to alignment column occupancy, which we define as the number of aligned residues in a column (i.e., $n_{seqs}$ - $n_{gaps}$).

### Regression model of column missense counts
Linear regressions of Pfam domain multiple sequence alignment column missense totals ($\Sigma X_{missense}$) against evolutionary divergence ($V_{Shenkin}$) were calculated with the R *lm* function and model equation:

$$\Sigma X_{missense} = \beta_1 n_{human} + \beta_{12} n_{human} * V_{Shenkin}$$

This model treats the evolutionary divergence as a modifier of the rate of missense variation per human residue. Independent regressions were calculated for each Pfam domain and parameter p-values were adjusted with Benjamini & Hochberg's FDR method. $\beta_1$ represents the average number of missense variants per human residue in the Pfam domain whereas $\beta_{12}$ is a measure of the sensitivity of this rate to the evolutionary divergence. A positive $\beta_{12}$ indicates that there are more missense alleles per site at divergent positions than there are at conserved positions; only positive $\beta_{12}$ is observed when this parameter is significant (Fig. 1e).

### The missense enrichment score (MES)
Columns were classified as depleted, enriched or neutral with respect to the missense variants in the column relative to the average for the other columns in the alignment. For each alignment column $x$, a $2 \times 2$ table was constructed of the form $a$, $b$, $c$, $d$ with elements: $a$. the number of variants mapped to residues in column $x$, $b$. the total number of variants mapped to all other alignment columns, $c$. the number of human residues in column $x$ and $d$. the total number of human residues in the rest of the alignment. The R *stats* function *fisher.test* (two-sided Fisher's exact test) yielded odds ratios OR > 1 if the column contained more than the alignment average number of variants per human residue or OR < 1 if there were fewer than the average number of variants per human residue. For a specified *threshold* columns with $p \geq p_{threshold}$ were defined as missense netural and columns with $p < p_{threshold}$ were defined as missense depleted if OR < 1 or enriched if OR > 1. In addition to the effect size, $p$ is sensitive to data availability (i.e., variant counts) and alignment column occupancy.

In this analysis, we defined missense depleted and missense enriched sites with a p-value threshold of 0.1. This allowed sensitive capture of both extremes, while acknowledging a higher tolerance of false positive classifications. In our two-tailed test, this threshold yields the same critical regions for each extreme as one-sided tests with $p < 0.05$, highlighting how our standard for significance is aligned with this convention. Importantly, the collection of tests across the sites in an alignment does not warrant multiple test correction due to the inherent interdependency of each result. For instance, in a simplified scenario with two sites, the MES odds ratios are reciprocal and the $p$ values identical. This diverges from the assumptions of standard test corrections, which require independent or at most weakly dependent tests. Finally, beyond the statistical justification, the clear biological relevance of our scheme underlines its effectiveness. Sites identified as missense depleted are consistently enriched in structural features and pathogenic variants, providing strong evidence that MES captures real biological effects.

### Determining conserved and divergent sites for comparison to missense depleted and enriched sites
The MES automatically corrects for domain-specific observations. In order to represent conventional conservation scores as calculated by Shenkin in a similar domain-specific way, we took the N most evolutionarily conserved columns in each family where N is the number of missense depleted positions in that family (i.e., MES < 1 and MES $p < 0.1$). We considered this a better approach than setting fixed Shenkin thresholds for the comparison to missense depletion because MES is the measure of the constraint at a position relative to the rest of the domain in contrast to Shenkin, which is an

absolute measure with its range significantly influenced by the depth of the alignment.

## Mapping structural features to Pfam alignments

Residue relative solvent accessibilities (RSAs) were calculated from the DSSP accessible surface[67] as described in Tien et al.[68] and were classified as surface (RSA ≥ 25%), partially exposed (5% ≤ RSA < 25%) or core (RSA < 5%). To differentiate residues that are buried within a single biopolymer from those buried by interacting molecules (e.g., small-molecule ligands, bound DNA, other proteins, etc.) RSA calculations were performed on each PDB chain separately after stripping all nonprotein atoms.

Interatomic interactions were calculated with Arpeggio[69] using defaults. A residue was considered to participate in a domain interaction if it interacted with a Pfam domain on a different PDB chain. Ligand-protein interactions were filtered by BioLip[70] to identify the most biologically relevant contacts.

Solvent accessibility and residue interactions were mapped to Pfam alignments via UniProt-PDB cross-references in SIFTS[71].

## Aggregating structural features over Pfam Domains

In this work, we aggregated data from all available PDB structures for all sequences in each alignment. This meant that any given sequence could have many, one or zero-mapped PDB structures. Since there may be more than one PDB structure for a sequence, in order to obtain Pfam domain-level summary statistics that are unbiased by the PDB coverage, we first summarised the structure features over each sequence before calculating alignment level statistics. For example, if a sequence has 20 3D structures and a specific residue interacts with a ligand in 5 of the structures, then we count this as only one interaction at the alignment level. This ensures that the Pfam-level structural classifications are representative of the sequence distribution of properties rather than skewed towards the properties of the sequences with the most PDB structures.

The PDB coverage of a residue is the number of PDB chains that cover that residue. The PDB total coverage of an alignment column would be the sum of the residue PDB coverages. The PDB sequence coverage of a column is the number of residues with PDB coverage of at least one.

## Pfam consensus solvent accessibility class

The solvent exposure class of an alignment column was defined as the most frequent exposure class (buried, part-exposed, surface) of the residues in the column, where the residue exposure class was defined as the most frequent exposure class calculated for that residue across all mapped PDB chains.

## Conservation plane categories

Columns were classified as conserved or unconserved with respect to residue diversity indicated by the Shenkin entropy across all sequences in the alignment relative to the other columns (i.e., including non-human sequences). This was achieved by taking the percentage rank of the Shenkin entropy for each alignment column within each family. Unconserved residues are defined as those amongst the 50% most divergent residues in the Pfam domain (i.e., Shenkin percent-rank ≥ 50%) and conserved residues are those that are less divergent than these (i.e., Shenkin percent rank < 50%). This constant rank threshold is effectively the same as setting a custom Shenkin entropy threshold for each Pfam and is simpler than our previous definition (above), which effectively set a different rank threshold per family too. The reason for this simpler approach is that we are no longer looking to compare the properties of population-constrained positions to evolutionarily conserved ones but to see how they complement each other.

Cross-classification by the missense enrichment score and conservation classes yields four distinct classes. These are unconserved-missense depleted (UMD), unconserved-missense enriched (UME), conserved-missense depleted (CMD) and conserved-missense enriched (CME). Columns where $p \geq 0.1$ are classed as conserved-missense neutral (CMN) or unconserved missense-neutral (UMN). See Fig. 3a.

## Feature enrichment tests

Where we report that a feature is enriched in a particular class of residues (e.g., protein-ligand interactions within missense-depleted positions) we quote the $p$ value from the two-sided Fisher's exact test.

## Statistics and reproducibility

Statistical analyses were conducted in R as outlined in detail in the preceding methods subsections and may be reproduced by running the R markdown notebooks available as specified in the Code Availability section of this manuscript.

## Reporting summary

Further information on research design is available in the Nature Portfolio Reporting Summary linked to this article.

## Data availability

This study employed the following publicly accessible datasets: Pfam31.0 (https://ftp.ebi.ac.uk/pub/databases/Pfam/releases/Pfam31.0/Pfam-A.full.gz), gnomAD v2 (https://gnomad.broadinstitute.org/downloads#v2), ClinVar (https://ftp.ncbi.nlm.nih.gov/pub/clinvar/vcf_GRCh37/archive_2.0/2018/clinvar_20180401.vcf.gz) and the Protein Data Bank (https://www.ebi.ac.uk/pdbe/). The aggregated datasets generated during this study, encompassing integrated analyses of protein families, genetic variants, and structural data are available from BioStudies repository S-BSST1137.

## Code availability

The code supporting this work is openly available from GitHub. R markdown notebooks used to produce the statistical results and figures in this manuscript can be accessed at https://github.com/bartongroup/SM_Pfam-gnomAD-statistics[72], and provide numerical source data for graphs and charts. Python packages developed for aggregating variant and structural data over multiple sequence alignments are available at the following GitHub repositories: https://github.com/bartongroup/SM_VarAlign[73], https://github.com/bartongroup/ProIntVar[74] and https://github.com/bartongroup/ProteoFAV[75]. The specific versions of these software tools used in our study have been archived and are available in Zenodo, with the DOI identifiers provided in the References section of this manuscript.

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

## Acknowledgements

We thank Drs Chris Cole and Melanie Volk who contributed to early ideas in previous versions of this work and Dr Marek Gierlinski for his advice on statistics. We also thank the University of Dundee High-Performance Computing Team for supporting the computing infrastructure on which this research was carried out as well as Dr Jim Procter and the Jalview development team for discussion of alignment/structure data visualisation. This work was supported by grants to GJB from UKRI-Biotechnology and Biological Sciences Research Grants [BB/J019364/1; BB/R014752/1] and Wellcome Trust Biomedical Resource Grants [101651/Z/13/Z; 218259/Z/19/Z]. FM was supported by Wellcome Trust Doctoral Training Account [100150/Z/12/Z], TBB was supported by Coordenação de Aperfeiçoamento de Pessoal de Nível Superior [CAPES process; 1529/12-9].

## Author contributions

S.A.M., F.M.M., T.B.B. and G.J.B. conceived designed and developed the research. S.A.M. analysed the data. S.A.M., F.M.M. and T.B.B. developed the software. S.A.M. and G.J.B. wrote, reviewed and edited the manuscript. G.J.B. secured funding and supervised.

## Competing interests

The authors declare no competing interests.
