## [Peer Review File · Communications Biology]

Reviewers' comments:

Reviewer #1 (Remarks to the Author):

This interesting paper combines information derived from conservation scores with information derived from SNPs observed in human genomes. The authors derive a combined score (MES) – Missense Enrichment Score - which classifies residue positions according to the number of variants seen in gNOMAD data (combining paralogues to the human sequence to increase the number of mutations observed for a given family). With these data, they go on to evaluate the odds ratios for whether mutations are enriched or depleted compared to the number of mutations seen at all positions in this particular family of proteins. This information is then used to categorise residue positions in proteins into 6 classes, depending on both the conservation score and MES score. They describe a ‘conservation plane’ – which is basically just a plot of MES vs conservation. They then include splitting according to RSA (relative solvent accessibility) to explore whether the conservation score just mirrors the MES score or whether they are different and thus allow a finer grained classification of residue positions than revealed by just considering conservation. The analysis depends on the odds ratios, which can be difficult to compare, given all the factors that contribute. Lastly they inspect 3 protein families (GPCRs, P450s and LBD of nuclear receptors), which have many human paralogues and have been widely studied, to see how the ‘conservation plane’ can distinguish those residue positions which are functional in all members of the family, from those that vary in different members and could be described as SDPs.

This is quite a long, detailed exposition of conservation & mutations, with a lot of effort to understand the observations and provide a rationale explanation. The quality of the results, depends on having a rather deep alignment and also knowledge of the structures, since results vary with solvent accessibility. Results are presented for a limited subset of Pfam families,

I think the paper could be much shorter and more concise (with less discussion on reasons and more concise methods); is there a web site to run a given Pfam family to obtain MES and conservation scores and thus identify SDPs? Some of the fig legends could be clearer (see below). The methods are quite clearly explained, but discussion is rather long and rambling.

Detailed comments:

Abstract: No mention of MES and 'the conservation plane', which to me are central to this paper.

L42 Method to identify SDPs seems qualitative rather than quantitative? Can you be more specific?

L 168 Given the importance of MES to paper, I think a fuller description in this part of the text explaining why this formulation was used would be helpful

Fig. 2 Confusion in colours in plots. Those in A are different than those in B-D. Use different colours if possible.

L203 – pls explain the statement – “MES is consistently more sensitive to exposure than Shenkin”

L223 The following discussion (in paragraphs) is quite difficult to follow and somewhat convoluted. Differences between MES and conservation are generally small, so perhaps some of this is being over-interpreted? Little mention is made of standard deviations, which I assume are represented in the plots. Need to consider what is the basic difference between MES and conservation – in time scales; species; lack of selection for an individual human; population effects in humans. This is discussed a little at the end, but is basic to all the discussion of the different classes. Maybe these principles could be introduced in more detail in the Introduction, to avoid repetition in the results section.

L231 What is 95% CI?

L242 Not sure this logic is correct? The conservation scores will include homologues binding different ligands, but so do the polymorphisms observed in the human paralogues? Did you make any attempt to differentiate functions of homologues and human paralogues??

L249 . Finally, we note that the greater discrimination at surface sites overall may reflect the fact that exposed residues are less constrained by packing and folding so that the presence of a binding site is a more likely explanation of a constrained surface (cf. a core position where constraint due to residue packing dominates). - Surely this is the most likely explanation? – why leave to last?

L347 Fig 3B - What determines the shape/disposition of rectangles in this area plot?

L 380 Fig. 4 – Different y-scales is misleading

L425 - Define PLIs

L474 – It is not clear to me why you need a priori knowledge before applying this approach? Pls clarify

L462 – Application to individual protein families.

This long and detailed section is really not easy to follow and I think it could be simplified to pick out a few principles, rather than trying to go into exquisite detail for each family and many residue positions. I must confess I gave up trying to wade through this quagmire. The figures are also detailed, but not easy to interpret. Linking the figures more closely to the text and only including the most important parts might help.

L634 Discussion

Since ref 84 was published in 1991, I would have expected to see this discussion in the Introduction – not right at the end. As suggested above, the ‘theoretical background to variants within species and conservation between species’ is at the heart of this study – and could be addressed generally in the introduction perhaps?

L694 – key application? This paragraph highlights the possibility of using these data to identify SDPs. What exactly is the protocol for doing this? It could be highlighted here and described simply. Is there a website for this?

L707 MES considerations – the Arg example seems to undermine the ‘enrichment/depletion’ concept across a whole protein. It is not clear to what extent this is a problem. It would be interesting to know for example how many of the ME residues are arginine. In contrast using the average mutation rate across a protein family seems like a reasonable way to identify critical residues therein.

IN summary, I certain support publication of this paper, but I would ask the authors to make it easier for the reader, but cutting out some of the detailed disucsions and highlighting key principles throughout.

Reviewer #2 (Remarks to the Author):

In this manuscript, the authors develop a novel measure, called MES, which is based on the enrichment extent of population variants appearing in the Pfam domain, to complement the traditional evolutionary conservation measure. Based on these two measures, the authors investigate the structural and functional attributes of different classes of residues, such as unconserved missense enriched (UME) and conserved missense enriched (CME) positions. The analyses may be meaningful and provide new insight into the study of missense mutations. I have the following comments that may be useful in improving the manuscript.

This is a general suggestion. Authors include all their findings in their manuscript, but the readers will have trouble finding the take-home message. I highly recommend that the authors enhance some core findings and remove minor findings in this manuscript.

Although authors perform a comprehensive analysis, it may be difficult for readers to easily acquire a big picture of this study. Thus, the authors could provide a schematic diagram that includes the major analyses and corresponding methods to facilitate the understanding of their work.

Based on the description of the first section in Methods (lines 783-788), readers (at least for me) may not understand the detailed process for mapping variants to the sequence in alignments. However, the mapping step is the basis of this research. Could the author provide examples for this step? Fig 1A only shows the final annotations. The annotation process could also be presented in the supporting file.

More details about the dataset could be provided. For instance, the proportion of human protein sequences and the proportion of missense variant positions in each domain could be summarized in figures.

Regarding the MES score, the threshold of p-value is 0.1 rather than 0.05 (the widely used value). Please provide reasons for choosing this value.

In this work, the major structural constraints of population variants are their solvent accessibilities. Why not consider other structural information, such as secondary structure states, and order/disorder states? Is it necessary to talk about these factors in the discussion section?

AACons offers 18 different conservation scores (line 798), among which the Shenkin score based on identity is selected for subsequent analyses. The rationale behind choosing Shenkin should be provided. When other conservation measures are selected, I wonder if the similar observations could be gained in this work.

In the abstract, the authors suggest that they develop a new method for predicting specificity determining positions (SDPs) and investigate SDPs in two protein families. However, the definition or explanation of SDP is missing in the main text.

The sentence presented in Lines 57-61 is too long, and there are no punctuation marks. Please rephrase this sentence.

We are grateful for the detailed, positive and constructive feedback provided by the reviewers, which has been helpful in refining our manuscript. Both reviewers were supportive of publication but highlighted the need to reduce length and increase clarity to focus on the main points of the research. The core of the revised manuscript (excluding Methods and References) is substantially reduced in length from 9,514 to 4,235 words while also answering the other suggestions of the referees.

Where relevant we have indicated relevant line numbers in the revised text which refer to the version with no tracked changes.

Reviewer #1 (Remarks to the Author):

This interesting paper combines information derived from conservation scores with information derived from SNPs observed in human genomes. The authors derive a combined score (MES) – Missense Enrichment Score - which classifies residue positions according to the number of variants seen in gNOMAD data (combining paralogues to the human sequence to increase the number of mutations observed for a given family). With these data, they go on to evaluate the odds ratios for whether mutations are enriched or depleted compared to the number of mutations seen at all positions in this particular family of proteins. This information is then used to categorise residue positions in proteins into 6 classes, depending on both the conservation score and MES score. They describe a ‘conservation plane’ – which is basically just a plot of MES vs conservation. They then include splitting according to RSA (relative solvent accessibility) to explore whether the conservation score just mirrors the MES score or whether they are different and thus allow a finer grained classification of residue positions than revealed by just considering conservation. The analysis depends on the odds ratios, which can be difficult to compare, given all the factors that contribute. Lastly they inspect 3 protein families (GPCRs, P450s and LBD of nuclear receptors), which have many human paralogues and have been widely studied, to see how the ‘conservation plane’ can distinguish those residue positions which are functional in all members of the family, from those that vary in different members and could be described as SDPs.

This is quite a long, detailed exposition of conservation & mutations, with a lot of effort to understand the observations and provide a rationale explanation. The quality of the results, depends on having a rather deep alignment and also knowledge of the structures, since results vary with solvent accessibility. Results are presented for a limited subset of Pfam families,

I think the paper could be much shorter and more concise (with less discussion on reasons and more concise methods); is there a web site to run a given Pfam family to obtain MES and conservation scores and thus identify SDPs? Some of the fig legends could be clearer (see below). The methods are quite clearly explained, but discussion is rather long and rambling.

We thank the reviewer for their helpful and candid feedback. We appreciate the recognition of our efforts to understand and rationalise our results and we have attempted to retain this quality while addressing the need to be more concise. We hope we have been able to balance this in an effective manner.

We have released Python packages that enable others to run our analysis on their own alignments on GitHub. We have also provided all our calculated data, including MES and conservation scores in BioStudies, which allows others to look up results for their own proteins, without the need to run the full pipeline. Links are provided in the Data and Code Availability sections. We have recently received new funding (UKRI-BBSRC) to support the development of a web resource based on this work which we plan to release in the coming months.

Detailed comments:

Abstract: No mention of MES and 'the conservation plane', which to me are central to this paper.

We have rewritten the abstract and made sure to emphasise these developments.

L42 Method to identify SDPs seems qualitative rather than quantitative? Can you be more specific?

We have been more precise in the revised abstract and now say explicitly that we find functional residues at unconserved sites, which can mean they are related to specificity. We have stated in several places in the revised manuscript that we suggest UMD sites are potentially related to specificity and why (L266, L377 and L394).

L 168 Given the importance of MES to paper, I think a fuller description in this part of the text explaining why this formulation was used would be helpful

Thank you for this suggestion, we agree that this is essential and have described the key benefits of MES in this section. Please see L164 in revised manuscript.

Fig. 2 Confusion in colours in plots. Those in A are different than those in B-D. Use different colours if possible.

We recognise how applying the same colour scheme to these different plots could be confusing. In our revision of the manuscript, we have removed Figure 2A to Supplementary Figure 2 to avoid this clash. We think this is a good solution since we realised this figure was redundant with the association statistics described in the text (L187) and the additional detail provided by this visualisation was a side note to our main results.

L203 – pls explain the statement – “MES is consistently more sensitive to exposure than Shenkin”

The crux of this matter is that our presentation of the results in this section was overly detailed and difficult to follow, as the reviewer rightly points out in the next comment. We hope our revision of this section corrects this problem and that our meaning and intentions are now clear and unambiguous. In our revised manuscript, all we highlight with respect to MES and exposure is that they are strongly associated, with missense depleted residues predominantly buried (revised L187).

L223 The following discussion (in paragraphs) is quite difficult to follow and somewhat convoluted. Differences between MES and conservation are generally small, so perhaps some of this is being over-interpreted? Little mention is made of standard deviations, which I assume are represented in the plots. Need to consider what is the basic difference between MES and conservation – in time scales; species; lack of selection for an individual human; population effects in humans. This is discussed a little at the end, but is basic to all the discussion of the different classes. Maybe these principles could be introduced in more detail in the Introduction, to avoid repetition in the results section.

We thank the reviewer for highlighting this section and agree it was difficult to follow. In our revision, we refocused this section on the main result that missense depleted positions are functionally enriched, and we have removed the detailed discussion of small quantitative differences between missense depleted and evolutionarily conserved sites.

Even though these differences were beyond the statistical error margins, we recognise that our rationalisation of these small effects were potential over-interpretations. We also realised that the lengthy discussion regarding these small points would detract emphasis from the main result and so agree it is better to cut these details out while continuing to provide the data for readers to make their own interpretations.

We are also grateful for the suggestion to include more theoretical background to evolutionary and population variation and have added a new paragraph to the introduction to this effect.

L231 What is 95% CI?

This referred to the 95 % confidence interval. We have defined this standard statistical abbreviation in the revised text.

L242 Not sure this logic is correct? The conservation scores will include homologues binding different ligands, but so do the polymorphisms observed in the human paralogues? Did you make any attempt to differentiate functions of homologues and human paralogues??

For reference, we said, "... if homologs across a domain family bind different ligands... we do not expect evolutionary conservation to be an ideal predictor of the binding residues but in contrast, MES will remain sensitive to these residues because it is calculated from population polymorphs."

What we meant was that individual polymorphs of each human paralog binds the same ligand. For example, testosterone receptors in different human individuals are constrained to bind testosterone, oestrogen receptors across individuals are similarly constrained, etc. So even though we are aggregating over the paralogs, the signal reflects population constraint across human individuals at each individual receptor. If all the paralogs bind their target at a homologous site, we can resolve this over the paralogs. This contrasts with conservation scores, which would reflect the diversity across homologs at a site where the residue varies to accommodate different ligands.

In our revision of the manuscript, we have removed these comments since the general idea is more clearly conveyed and evidenced in the context of our novel "UMD" residue classification (L268).

L249 . Finally, we note that the greater discrimination at surface sites overall may reflect the fact that exposed residues are less constrained by packing and folding so that the presence of a binding site is a more likely explanation of a constrained surface (cf. a core position where constraint due to residue packing dominates). - Surely this is the most likely explanation? – why leave to last?

We agree with the reviewer's assessment and have made this point more prominent. Thank you for raising this issue. Please see L193 in revised manuscript.

L347 Fig 3B - What determines the shape/disposition of rectangles in this area plot?

The rectangles in the plot are laid out so that each class corresponds with its general position in panel A. We now clarify this in the Figure legend and that only the relative areas portray quantitative data.

L 380 Fig. 4 – Different y-scales is misleading

We acknowledge the reviewer's concern that y-scales can be manipulated to mislead; however this is not the case here. We have simplified the new version of this figure as Figure 3 with details consigned to Supplementary Figure 4. Figure 3 D, E and F refer to different enrichments (ligand binding, protein interface and pathogenic variants) that are naturally on a different scale.

We did not intend this figure to highlight the different effect sizes observed for ligand binding, protein binding and pathogenic variants, which would require a constant y-scale across all three panels and would conceal the main point that protein interacting residues are differentially distributed between CMD, UMD etc. In all plots, 95 % confidence intervals are shown for all points, providing further transparency.

L425 - Define PLIs

This was an abbreviation for protein-ligand interactions. In our revision, we have avoided the need to use the abbreviation.

L474 – It is not clear to me why you need a priori knowledge before applying this approach? Pls clarify

As part of this reviewer's recommendation below we have substantially simplified the discussion of examples and this section has now been deleted.

L462 – Application to individual protein families.

This long and detailed section is really not easy to follow and I think it could be simplified to pick out a few principles, rather than trying to go into exquisite detail for each family and many residue positions. I must confess I gave up trying to wade through this quagmire. The figures are also detailed, but not easy to interpret. Linking the figures more closely to the text and only including the most important parts might help.

We thank the reviewer for their suggestion to substantially cut this section. We have removed the detailed analysis of the individual families and incorporated the key principles and observations into the results sections describing UMD and CMD sites. We have selected the most important panels from across the original figures and combined these into a single new figure.

L634 Discussion

Since ref 84 was published in 1991, I would have expected to see this discussion in the Introduction – not right at the end. As suggested above, the 'theoretical background to variants within species and conservation between species' is at the heart of this study – and could be addressed generally in the introduction perhaps?

We have included the McDonald-Kreitman test in the introduction as suggested (L94)

.

L694 – key application? This paragraph highlights the possibility of using these data to identify SDPs. What exactly is the protocol for doing this? It could be highlighted here and described simply. Is there a website for this?

Our idea is that UMD sites are candidate specificity determining positions. We have clarified that we mean this in several places in the revised manuscript (L266, L377 and L394).

In practice, we would recommend considering the UMD assignment in context with protein structure data aggregated across the family, in the way we described for GPCRs and nuclear receptors.

We are currently developing a web server to facilitate this sort of analysis, but at present it is possible to obtain UMD assignments for Pfam from our BioStudies repository, or alternatively our pipeline can be run locally by installing our Python package from GitHub.

L707 MES considerations – the Arg example seems to undermine the ‘enrichment/depletion’ concept across a whole protein. It is not clear to what extent this is a problem. It would be interesting to know for example how many of the ME residues are arginine. In contrast using the average mutation rate across a protein family seems like a reasonable way to identify critical residues therein.

We appreciate the reviewer's recognition of the suitability of the average mutation rate across protein families as a baseline for our Missense Enrichment Score (MES), particularly to identify critical sites. This was one of our core design considerations and is validated by the diverse structural and functional associations with MES, providing strong support that the calculation captures the effects of these constraints.

Addressing the hypermutability of Arginine, we raised this in our discussion because we knew that it is a significant factor, but we do not think it compromises the overarching concept of missense enrichment or depletion across proteins. Instead, it underscores the intricate relationship between genetic constraint and mutability in our analysis. We did consider adjustments for residue variation rates. However, our current focus was on how missense variant distributions in protein domains relates to protein structure, interactions and pathogenicity. While undoubtedly important, full consideration of residue biases introduce a level of complexity that was not required for our primary objective. Our aim with MES is to establish an effective constraint score, which through validation against structure and pathogenicity, it seems clear we have achieved.

In summary, while the hypermutability of Arg and other residue-specific biases are interesting and merit further exploration, they do not undermine the validity of MES. The broader implications of our findings, particularly in understanding protein structure and function, are of primary relevance to our current study.

IN summary, I certain support publication of this paper, but I would ask the authors to make it easier for the reader, but cutting out some of the detailed disucssions

We sincerely thank the reviewer for their support and hope we have improved the manuscript in line with their expectations. Their advice to cut down detailed discussions and emphasise key principles has been our core objective in this revision of the manuscript, and we are optimistic that this has substantially improved our report.

Reviewer #2 (Remarks to the Author):

In this manuscript, the authors develop a novel measure, called MES, which is based on the

enrichment extent of population variants appearing in the Pfam domain, to complement the traditional evolutionary conservation measure. Based on these two measures, the authors investigate the structural and functional attributes of different classes of residues, such as unconserved missense enriched (UME) and conserved missense enriched (CME) positions. The analyses may be meaningful and provide new insight into the study of missense mutations. I have the following comments that may be useful in improving the manuscript.

This is a general suggestion. Authors include all their findings in their manuscript, but the readers will have trouble finding the take-home message. I highly recommend that the authors enhance some core findings and remove minor findings in this manuscript.

We thank the reviewer for their recognition of the relevance of our work to the study of missense mutations and are grateful for their advice to focus our narrative. We have taken to heart their criticism that our attempt to include and comment on every result is counterproductive to raising the understanding of our work. In our revision, we have followed the reviewer’s advice to emphasise core results and remove minor findings as best we can and believe that this has substantially improved the communication of our work.

Although authors perform a comprehensive analysis, it may be difficult for readers to easily acquire a big picture of this study. Thus, the authors could provide a schematic diagram that includes the major analyses and corresponding methods to facilitate the understanding of their work.

We hope that following the substantial revision of our text that it is much easier to understand our overall approach and our main results, so that this should no longer be required.

Based on the description of the first section in Methods (lines 783-788), readers (at least for me) may not understand the detailed process for mapping variants to the sequence in alignments. However, the mapping step is the basis of this research. Could the author provide examples for this step? Fig 1A only shows the final annotations. The annotation process could also be presented in the supporting file.

We have added the figure below to the supplementary material and signposted this in Methods.

Supplementary Figure 5. Schematic illustrating how missense variants from gnomAD are mapped to Pfam domain multiple sequence alignments. An excerpt of a Pfam domain family alignment is shown where sites with missense variants in

gnomAD are highlighted red. The tables below the alignments show the UniProt residue mappings that are extracted from the Pfam Stockholm alignment files and variant records from the *gnomAD* VCF with Ensembl VEP annotations. The merge keys between the tables are indicated by the dashed lines.

More details about the dataset could be provided. For instance, the proportion of human protein sequences and the proportion of missense variant positions in each domain could be summarized in figures.

We have added plots to show these parameters in Supplementary Figure 1, copied below.

Supplementary Figure 1. Additional plots showing the distribution of human sequences and missense variants in our Pfam-*gnomAD* dataset. A. Number of sequences (all species) in Pfams with at least one human sequence. B. Number of human sequences in these Pfam domains. C. Number of human sequences vs. total number of sequences. D. Number of missense variants vs. number of human sequences in Pfam domains. E. Histogram of the ratio of missense variants to human residues. F. The missense residue ratio vs. the number of human residues in the domain.

Regarding the MES score, the threshold of p-value is 0.1 rather than 0.05 (the widely used value). Please provide reasons for choosing this value.

We chose 0.1 after exploring alternative thresholds including 0.01, 0.05 and lower as a good balance of the need to retain enough sites for analysis whilst minimising false positives.

It is also relevant to note that the $p < 0.1$ threshold for two-tailed tests, as we use here to detect high or low extremes of missense variation at a site, yields the same critical regions for each extreme as a one-tailed test with $p < 0.05$ does. This means we are essentially using a similar standard for determining significance in terms of the extremity of the observed results under the null hypothesis.

Finally, the sites we define as missense depleted are shown to be consistently enriched in structural features and pathogenic variants, as we would expect for constrained sites, which lends further confidence to our classification.

We have added further technical notes on MES p-value to the MES section in methods to explain our decisions more clearly in this regard (L479).

In this work, the major structural constraints of population variants are their solvent accessibilities. Why not consider other structural information, such as secondary structure states, and order/disorder states? Is it necessary to talk about these factors in the discussion section?

This is a good question. Integrating secondary structure and disorder would be an interesting addition to our work. However, solvent accessibility was the only essential consideration because it strongly confounds interactions, especially protein-protein interactions, which we were keen to focus on. Additionally, exposure has well-established associations with evolutionary substitution rates and pathogenic variants, so this provided a good starting point for the analysis.

An interesting consideration is that patterns in residue exposure often result from secondary structure, and this is often what drives secondary structure related constraint, so this element will be somewhat captured by our analysis. An analysis of disordered regions might be particularly interesting since population constraint would be a novel way to help characterise residues within them.

While these points are interesting, we don't think they need to be addressed in our discussion, especially with our efforts to focus on only the main points in our broad study.

AACons offers 18 different conservation scores (line 798), among which the Shenkin score based on identity is selected for subsequent analyses. The rationale behind choosing Shenkin should be provided. When other conservation measures are selected, I wonder if the similar observations could be gained in this work.

In our initial analyses we found significant correlation between the different conservation scores, particularly among scores belonging to the same category, such as assessing symbol diversity (including Shenkin) or considering physicochemical properties. For this analysis, we opted for Shenkin owing to its simple interpretation of amino acid diversity at each site, which we thought was a good match for our similarly interpretable MES formulation.

In our analysis we include results that show Shenkin on its own is sensitive to exposure, interactions and pathogenic sites (Figure 2), so it is clear that it captures evolutionary constraint across diverse families in an intuitive manner. We also found Shenkin to be a useful reflection of conservation in an alignment in previous detailed work on Ankyrins.

We have added a few sentences to explain our selection of the Shenkin score to the represent evolutionary conservation in the methods (L438).

In the abstract, the authors suggest that they develop a new method for predicting specificity determining positions (SDPs) and investigate SDPs in two protein families. However, the definition or explanation of SDP is missing in the main text.

Thank you for highlighting this oversight. We have provided an explicit definition on L269 and also revisit this in the discussion of SDPFox (L394) of the revised manuscript.

The sentence presented in Lines 57-61 is too long, and there are no punctuation marks. Please rephrase this sentence.

Thank you for highlighting this, we have rephrased this sentence with to use more concise language and better organisation.

We thank Reviewer #2 for their detailed feedback and questions, and we hope we have responded to these queries to their satisfaction.

REVIEWERS' COMMENTS:

Reviewer #2 (Remarks to the Author):

The authors have addressed my concerns. Thank you!

We are grateful that the responding referees are satisfied with our changes and responses to their reviews and acknowledge we have no further requests or queries to answer. Accordingly, only formatting and editorially requested changes have been made to the resubmitted manuscript.

Reviewer #2 (Remarks to the Author):

The authors have addressed my concerns. Thank you!

We thank the reviewer for recognising our efforts to address their concerns and would like to thank them once more for their helpful comments.